# Laser-Induced Graphene Heater Pad for De-Icing

**DOI:** 10.3390/nano11113093

**Published:** 2021-11-16

**Authors:** Jun-Uk Lee, Chan-Woo Lee, Su-Chan Cho, Bo-Sung Shin

**Affiliations:** 1Department of Cogno-Mechatronics Engineering, Pusan National University, Pusan 46241, Korea; lju3534@naver.com (J.-U.L.); cwleeho2@naver.com (C.-W.L.); cho_brian@naver.com (S.-C.C.); 2Department of Optics and Mechatronics Engineering, Pusan National University, Pusan 46241, Korea

**Keywords:** flexible, laser-induced graphene, polyimide, heater

## Abstract

The replacement of electro-thermal material in heaters with lighter and easy-to-process materials has been extensively studied. In this study, we demonstrate that laser-induced graphene (LIG) patterns could be a good candidate for the electro-thermal pad. We fabricated LIG heaters with various thermal patterns on the commercial polyimide films according to laser scanning speed using an ultraviolet pulsed laser. We adopted laser direct writing (LDW) to irradiate on the substrates with computer-aided 2D CAD circuit data under ambient conditions. Our highly conductive and flexible heater was investigated by scanning electron microscopy, transmission electron microscopy, Raman spectroscopy, thermogravimetric analysis, X-ray photoelectron spectroscopy, X-ray diffraction, and Brunauer–Emmett–Teller. The influence of laser scanning speed was evaluated for electrical properties, thermal performance, and durability. Our LIG heater showed promising characteristics such as high porosity, light weight, and small thickness. Furthermore, they demonstrated a rapid response time, reaching equilibrium in less than 3 s, and achieved temperatures up to 190 °C using relatively low DC voltages of approximately 10 V. Our LIG heater can be utilized for human wearable thermal pads and ice protection for industrial applications.

## 1. Introduction

Flexible graphene-based electronics are rapidly gaining interest because graphene, a unique two-dimensional (2D) nanomaterial, has great potential in electronics owing to its high electrical conductivity, excellent mechanical flexibility, and optical and thermal properties [1,2,3,4]. The electrothermal effect of graphene-based materials has received much attention because it is expected that they will substitute heavy metal heater products in a variety of fields, especially in the aeronautics and automotive industries, for de-icing and icing protection [5,6,7]. However, the feasibility of graphene-based heaters has been impeded by challenges such as difficulty in matching the high efficiency of metal heaters and the complex fabrication processes of graphene, such as chemical vapor deposition (CVD) and wet-chemical approaches [8,9,10,11]. Graphene-based heaters generally include the deposition of graphene oxide (GO) for its reduction and the multiple transfer of graphene [12,13,14,15,16,17]. Although these graphene-based composites have sufficient properties for manufacturing heaters, these processes are limited to resolving the challenges of economically viable mass production.

In 2014, a promising alternative to printed graphene patterns, called laser-induced graphene (LIG), was presented by Tour et al. [18]. They irradiated an abundant carbon precursor substrate to fabricate LIG using inexpensive and accessible 10.6 μm ${\mathrm{CO}}_{2}$ lasers, which are commonly used for cutting and etching in industry. LIG is a one-step, lithography-free process upon which a laser converts $sp^{3}$-hybridized carbon found in polymeric substrates into $sp^{2}$-hybridized carbon, which is the carbon allotrope found in graphene. LIG is a versatile technique that has been used to produce 3D graphene that is superhydrophobic or hydrophilic, doped with metal oxide nanocrystals, functionalized with polymers, or developed into vertically aligned graphene fibers [19,20,21,22,23,24]. The utility of LIG has been demonstrated in numerous applications, including supercapacitors, non-biofouling surfaces, transparent electrodes, sensors, and triboelectric nanogenerators [25,26,27,28,29]. LIG has opened up a multitude of possibilities for the mass production of a wide range of novel, eco-friendly 3D graphene-based applications. The simplified architecture and facile fabrication of 3D graphene allows for the full integration with the emerging field of one-step printed electronics, because this facile LIG manufacturing protocol eliminates the need for ink preparation, ink printing, and post-print annealing processes associated with solution-phase printed circuits [30,31,32,33,34]. While promising graphene-printed flexible electronics have been demonstrated at a lab scale, there is limited research on LIG-based applications such as heaters and gas sensors using the electrothermal effect based on localized Joule heating [35,36,37,38].

Considering the immense potential of LIG and the limited research on its thermoelectric applications, we studied herein the electrothermal performance of LIG patterns fabricated by an ultraviolet (UV) pulsed laser instead of a continuous wavelength (CW) infrared (IR) laser with high laser power. We adopted laser direct writing (LDW) to irradiate the polyimide (PI) films with computational circuit patterns with 2D CAD data under ambient conditions. Conductive LIG patterns can be easily fabricated using LDW on demand for customized patterns. To ensure the advantages of LIG patterns, the prepared LIG heaters were evaluated in terms of their electrical properties, electrothermal performance, and mechanical durability. The LIG heater demonstrated a rapid response time, reaching equilibrium within 3 s and achieving temperatures up to 190 °C using relatively low DC voltages of approximately 10 V. Their outstanding characteristics were confirmed by applying them to human wearable thermal pads and ice protection for industrial applications.

## 2. Materials and Methods

### 2.1. Laser Delivery System

A commercial 355 nm pulsed laser system (AONano 355-5-30-V from Advanced Optowave, Ronkonkoma, NA, USA) was used as the laser source for fabricating the LIG. Our $\mathrm{Nd}:{\mathrm{YVO}}_{4}$ laser with a maximum average power of 5 W at λ = 355 nm, pulse length τ = 15 ns, and repetition rate ƒ = 30 kHz was used to irradiate the samples in a standard atmospheric environment (25 °C and normal pressure), as described in Appendix A. LDW was adopted for the preparation of LIG patterns during laser induction under ambient conditions at room temperature. Laser beam delivery was performed by moving the mirrors of the galvano scanner (HurrySCAN III 14 from SCANLAB, Pucheim, Germany) and the F-θ lens of focal length f = 105.9 mm (S4LFT4100/075 Telecentric Scan Lens from Sill Optics, Wendelstein, Germany). In general, four key parameters are expected to affect laser-induced carbonization: laser power, scanning speed, hatch distance, and laser mode of scanning. In our best conditions, we fixed the laser parameters in the configuration where laser power = 1.3 W, hatch distance = 0.1 mm, and laser scan mode was unidirectional.

### 2.2. Fabrication of Prepared LIG Heater by UV Pulsed Laser

Appendix A describes our laser beam conditions for the LIG patterns. PI film (Kapton^®^ HN) with 125 μm thickness from Dupont^TM^ (Wilmington, DE, USA) was used as received. Kapton has a significant optical transmittance in the 10.6 μm region, which results in a rather deep penetration in the polymer substrate [24]. This limits the minimum thickness of the Kapton substrate, either by the complete penetration of the polymer or by the buckling of the film owing to the photothermal process, leading to non-flat devices that could be an obstacle for mechanical durability [39]. However, because of the presence of chromophores (conjugated double bonds and multi-aromatic rings) in the Kapton films, there is a strong complicated absorption in the ultraviolet region [40]. Although the absorption rate varies depending on the fluence, it has an absorption rate of over 85% from a very low fluence of 10 mJ/cm^2^ at a 355 nm wavelength [41]. This interaction between PI and a UV-pulsed laser allows for both thinner and narrower devices. Appendix A shows that various types of laser-patterned LIG samples were successfully produced using our defocused laser beam conditions.

The fabrication process is illustrated schematically in Figure 1. As shown in Figure 1a, the laser beam was scanned with the cross-hatching process to form grooved LIG patterns. The hatch distance was fixed at 0.1 mm. Both the horizontal and vertical scan modes gave high roughness to the surface.

We attempted to find the best condition for a grooved LIG structure that has excellent electrothermal performance with a variance of dynamic fluence according to laser scanning speed. Figure 1b shows an illustration of irradiation onto the PI surface with a 355 nm pulsed laser. The laser was irradiated on the Kapton films according to the dynamic fluence with the laser beam condition, as shown in Appendix A, using (2) to determine the optimum conditions for LIG patterns with electrothermal performance. After LIG patterning, Cu tapes were attached to both ends of the LIG patterns with silver paste, as shown in Figure 1c. Figure 1d shows the Kapton tape and aluminum foil used for encapsulation of the LIG heater.

As shown in Figure 1e, a heater pad could be used to defrost the aircraft’s wings by joule heating. Figure 1f, g show actual photographs of the LIG patterns and wearable heater pad, respectively. They show good flexibility for human motion. As shown in Figure 1h,i, our LIG pattern had a unique structure. When observed from the top view, the shape of the pattern was maintained, but pores were formed inside. The intersection in Figure 1j also shows that the structure of the LIG pattern contains many pores. This unusual pattern has the effect of accumulating generated heat in the pores.

### 2.3. Characterization

Optical photographs of the LIG patterns were taken with an optical microscope (BX60M system OLYMPUS, Shinjuku, Japan). Field emission scanning electron microscopy (FE-SEM) of the LIG was conducted on a field emission scanning electron microscope (TESCAN MIRA 3 LMH In-Beam Detector, Brno, Czech Republic). Transmission electron microscopy (TEM) images were obtained using a JEM-2100F microscope (JEOL, Akishima, Tokyo). Raman spectra were measured using a Raman spectrometer (NRS-5100 JASCO International Co., Ltd., Tokyo, Japan) with a 532 nm excitation line. Fourier transform infrared spectroscopy (FTIR) spectra were obtained using an FTIR-4100 type A instrument with an ATR-PRO 450-S accessory (JASCO International Co., Ltd., Tokyo, Japan). To analyze the composition and chemical bond states of the LIG patterns, X-ray photoelectron spectroscopy (XPS) spectra were analyzed using a K-Alpha™ X-ray Photoelectron Spectrometer System (Thermo Scientific, Waltham, MA, USA). The composition and crystal structure of the prepared LIG samples were determined by X-ray diffraction (XRD) using an X’Pert-MPD System (PHILIPS, Amsterdam, The Netherlands) with Cu K-α radiation (λ = 1.54 Å). The instantaneous electrothermal performance was recorded using an LCR meter 4100 (Wanye Kerr Electronics, Woburn, MA, USA) and a Keithley 2450 source meter (Keithley Tektronix, Beaverton, OR, USA). The zeta potentials were analyzed using a Litesizer 500 (Anton Paar, Graz, Austria). The surface area and pore size were recorded using an Autosorb-iQ (Quantachrome, Boynton Beach, FL, USA). Thermogravimetric analyzer data were studied with a Pyris (Perkin Elmer, Waltham, MA, USA). The water-droplet contact angle were measured by using a contact system (SDC-350, SIN DIN Corporation, Chengdu, China).

### 2.4. Formulation

The domain size (La) and crystalline size (Lc) of the LIG can be calculated from the XRD characteristics using the following equations [18]:(1)$$L_{a}=\frac{1.84\lambda }{B\left(2\theta \right)\cos\theta }$$
(2)$$L_{c}=\frac{0.89\lambda }{B\left(2\theta \right)\cos\theta }$$
where B(2θ) (in radian units) is the full width at half maximum of peaks (002) and (100), and λ is the wavelength of the X-rays (λ = 1.54 Å).

The normalized Brunauer–Emmett–Teller (BET) surface can be defined as follows [42]:(3)$$\frac{\frac{P}{P_{0}}}{V\left(1-\frac{P}{P_{0}}\right)}=\frac{1}{V_{m}C}+\frac{C-1}{V_{m}}\times \frac{P}{P_{0}}$$
where *P*/$P_{0}$ is the relative pressure, $V_{m}$ is the volume of the adsorbed gas, and *C* is the BET constant, which is used to evaluate the change in the volume of adsorption gas according to the pressure change.

## 3. Results

### 3.1. Morphological Characterization

Figure 2 shows the FE-SEM images of the pulse-overlapped LIG patterns on the PI. Figure 2b–e show the patterns fabricated at a speed of 20 mm/s. The LIG surface has a porous structure with the pore sizes ranging from 5 to 8 μm, which is similar to the pattern fabricated by a ${\mathrm{CO}}_{2}$ laser with a fluence of 6.6 J/cm^2^ [43]. The porous surface structure is likely caused by the escape of gases such as CO and $H_{2}$, owing to the localized high temperature and pressure [44]. Figure 2f–i shows the patterns fabricated at a laser speed of 40 mm/s. Owing to the relatively lower dynamic fluence, the micropore structure on the surface disappeared. Most of the peeled surface layers disappeared, revealing a relatively clean surface. This means that the pore radius on the surface was much smaller at the nanometer scale with a lower pressure and temperature than the prepared samples fabricated at a laser scanning speed of 20 mm/s. Figure 2j–m show the patterns fabricated at a speed of 60 mm/s. At this laser speed, the LIG patterns showed delamination of the surface. We predicted that a relatively faster laser scanning speed would result in more photochemical ablation [45]. As shown in Figure 2l–m, the porosity appears to disappear from the LIG patterns. We assumed that when the laser scanning speed was 60 mm/s, the interaction between the PI and UV laser showed the most effective photo-thermal and chemical effects. Figure 2n–q show the patterns fabricated at a speed of 80 mm/s. Figure 2n,o show that there was a more pronounced delamination effect. A more peeled layer was formed on the surface, and it seems that the photochemical effect on the surface was more dominant than the photo-thermal effect. Figure 2r–u show the patterns fabricated at a laser speed of 100 mm/s. Figure 2b,c show that the PI surfaces have a lot of delaminated layers [45]. As shown in Figure 2t–u, the thickness and porosity of the LIG patterns greatly decreased owing to the relatively lower dynamic fluence.

### 3.2. Chemical Characterization

As indicated in Figure 3a, the Raman spectrum of the LIG patterns presented three characteristic peaks of graphenic carbon [46]. The D band, positioned at 1346 cm^−1^, is induced by aromatic carbon domains, and when these domains are larger than 2 nm, the peak intensity increases with the number of defects and edges. The G-band at 1580 cm^−1^ arises from the stretching of $sp^{2}$ carbon bonds in the graphitic materials. The narrowing of the bands implies a transition from amorphous to crystalline states. In addition, the peak intensity ratios of the $I_{G}$/$I_{D}$ bands are important indicators for understanding defect density. The defect density is related to photo-thermal graphitization, which indicates the transformation to graphene-like structures. As the laser scanning speed decreases, both the band and the indicator $I_{G}$/$I_{D}$ decrease under our laser conditions. The 2D peak at 2690 cm^−1^ is uniquely associated with two-dimensional graphite with randomly stacked graphene sheets along the c-axis. The stronger 2D peak suggests a more complete graphitization of PI and the formation of graphene. The 2D peak increases as the laser scanning speed decreases [47].

X-ray diffraction analysis was performed to evaluate the spacing between the layers on the LIG heaters. The XRD patterns at scanning speeds of 20, 40, 60, 80, and 100 mm/s are shown in Figure 3b. Three main peaks at all scanning speeds were deconvoluted at 15°, 22.3°, and 26°, corresponding to the (001), (002), and (002) planes, indicating that LIG heaters were successfully prepared and uniformly consisted of reduced graphene oxide (RGO). The peaks only show various intensities at 22.3°, the parameter for the degree of graphitization, depending on the scanning speed of the laser [18,48]. The typical peak at 14° was attributed to the formation of oxygen functional groups in the LIG heaters. This corresponds to a d-spacing of 6.3 Å, which is larger than that of pristine graphene owing to the existence of oxygen functional groups. The other characteristic peaks indicate the disordered structure and unoxidized carbonaceous materials in the LIG heater, corresponding to a d-spacing of 4 and 3.4 Å, respectively [18,48]. These results show that the average d-spacing agrees well with observed TEM images (Appendix A). The LIG heater with a scanning speed of 40 mm/s shows the intense peak at 22.3°, confirming the high degree of graphitization, whereas that with scanning speed of 80 mm/s shows the weak peak at 22.3°. Calculating the domain size (La) and crystalline size (Lc) from Equations (1) and (2), the corresponding values of Lc and La are 3.91 nm and 2 nm for the 15° peak, 3.76 nm and 1.92 nm for the 22.3° peak, and 6.08 nm and 3.11 nm for the 26° peak. As shown in Figure 3c–e, XPS was employed to understand the chemical bonds of the LIG heater. The ratio of the C1s peak to the O1s peak indicates the formation of oxygen functional groups [49]. The low ratio of the C1s peak to the O1s peak is attributed to the highly oxidized LIG patterns (Figure 3c). The heaters with scanning speeds of 20 mm/s and 40 mm/s display a relatively low ratio of the C1s peak to the O1s peak, suggesting an increased amount of oxygen functional groups. The C1s spectrum of the heater with a scanning speed of 60 mm/s is shown in Figure 3e, which contains three main peaks at 284 eV (C=C bonds), 286 eV (C–O bonds), and 289 eV (O–C=O bonds), respectively [50]. The intensity of the C–O bonds is relatively higher than that of the pristine graphene, and the heater is determined to be graphene oxide (GO) and can work as a Joule heater with applied voltage. FTIR analysis of LIG heaters is observed in Figure 3d, showing the absence of any intense graphite peaks. Distinguishable intensity peaks of the heaters were observed at 3325, 1720, 1632, 1180, and 1040 cm^−1^, corresponding to the stretching of phenolic hydroxyl groups (–OH), carbonyl and carboxyl groups (C=O), sp2 carbon groups (C=C), epoxide groups (C–O–C), and epoxy groups (C–O), respectively [51,52]. The analysis indicated the presence of GO with little difference depending on the scanning speed of the laser.

The Brunauer–Emmett–Teller (BET) surface area of the LIG heater was measured by N2 adsorption and desorption using a Quantachrome (Autosorb-iQ) from Florida, USA. The specific surface area and pore size of the material were analyzed by adsorbing N2 gas to the sample and measuring the adsorption amount by partial pressure. The normalized BET surface is defined by Equation (3). The specific surface areas for each LIG heater with BET analysis were 116.695 m^2^/g, 103.357 m^2^/g, 64.045 m^2^/g, 44.121 m^2^/g, and 15.861 m^2^/g, and the average pore diameters were 30.42 Å, 42.02 Å, 67.72 Å, 44.96 Å, and 69.86 Å, in order of ascending laser fluence. The results show that as the laser fluence increased, the BET surface area decreased (Figure 4a–d). It was found that a high scanning speed shows a significantly lower $V_{m}$ than the others, and the smaller the specific surface area, the higher the thermal conductivity. The major weight loss of the LIG heaters can be explained based on the decomposition temperature of the internal chemical structure. The thermogravimetric analysis (TGA) results suggest a change in weight of the sample according to the temperature change and show the thermal stability of the sample and the composition ratio of the materials (Figure 4e). In general, a weight loss at approximately 100 °C is caused by the evaporation of moisture from the sample [53,54]. Water molecules that are easily decomposed by heat exist as hydrogen bonds between the epoxy group and the hydroxyl group, and the weight loss that occurs at 100–400 °C can be associated with the removal of oxygen-containing functional groups. In this process, water molecules are removed in stages, resulting in weight loss at approximately 180 °C [55]. Thereafter, the carboxyl and carbonyl groups are gradually removed at a high temperature of 500 °C [56]. As shown in Figure 4e, TGA in air suggests that increasing the laser fluence improves the thermal stability of the LIG heater. Figure 4f shows the zeta potential distribution depending on the scanning speed of the laser, confirming the presence of negative charges on the surface of the heaters at all scanning speeds. The negative charges are ascribed to the ionization of the carboxylic acid and phenolic hydroxyl groups observed in the FTIR analysis of GO [57]. The LIG heater with a scanning speed of 60 mm/s contains a zeta potential of −0.3 mV, which is the lowest value among the scanning speeds ranging from 20 to 100 mm/s. Since the heaters with scanning speeds of 80 mm/s and 100 mm/s have imperfect GO structures, they carry negative charges, resulting in a more negative zeta potential than those with a scanning speed of 60 mm/s. The heaters with scanning speeds of 20, 40, and 60 mm/s consist of a 3D carbon network structure of GO, containing many ionizations of the carboxylic acid and phenolic hydroxyl groups, resulting in a negative zeta potential. Since the LIG heaters fabricated at a scanning speed of 60 mm/s contain the appropriate ionization of the carboxylic acid and phenolic hydroxyl groups, they display the lowest zeta potential. The result depending on the scanning speed of the laser agrees with the XRD and elemental analysis TGA, which reveal the existence of hydrogen and oxygen.

### 3.3. Electrical and Thermal Properties of LIG Pattern

The electrical properties of the LIG heaters were investigated for different laser scanning speeds. As displayed in Figure 5a, the I–V curves for the LIG heaters exhibited a linear relationship, indicating good ohmic behavior in all cases. The heater with a scanning speed of 40 mm/s shows the highest slope of the I–V curve because of its good conductance of the pattern. The cyclic bending method was used to observe the electromechanical durability. We measured the change in electrical resistance after every 100 bending cycles, as shown in Figure 5b. The resistance of the LIG patterns is mostly constant, except for the change in resistance of the LIG patterns fabricated at laser speeds of 80 mm/s and 100 mm/s. This means that the LIG patterns fabricated from the laser scanning speed of 20 to 60 mm/s have a stable, well-formed conductive carbon network structure [58]. The resistance change according to bending was performed using a cylinder model with various radius (r) values, and its impact on the heater is summarized in Figure 5c. By decreasing the radius of the cylinder, the resistance shift was measured for our LIG patterns. All the LIG patterns show that the resistance increases slightly as the radius of the cylinder decreases (or the curvature represented by 1/r increases). The excellent characteristic of the flexibility of the patterns is that they maintain the electrical characteristics after a bending test. The LIG patterns fabricated from laser scanning speeds of 20 mm/s and 40 mm/s showed a flat slope, which indicates good flexibility [59].

The thermal performance studies obtained for the LIG heaters with an area of 100 mm^2^ are shown in Figure 5d. As shown in Appendix A, it was found that the size of each pattern and resistance between the electrode and LIG patterns are important factors for the heater. Our LIG heaters offer a quick response to the voltage loads, and the LIG heater fabricated at 80 mm/s can reach approximately 200 °C under 12 V within a very short period of time (less than 1.5 s)—this is the best thermal performance as far as we are aware [12,13,14,15,16,17,37,60]. The temperature changes depending on the power density of all the LIG-heaters are shown in Figure 5e. When the voltage was increased, all LIG heaters demonstrated an almost monotonous linear behavior [12]. The power densities to reach the saturation temperature of each heater are 20 mm/s (1.2 W/cm^2^ at 5 V), 40 mm/s (2.05 W/cm^2^ at 5 V), 60 mm/s (1.2 W/cm^2^ at 6 V), and 80 mm/s (1.05 W/cm^2^ at 7 V). Therefore, our LIG heater can provide not only a quick response to 200 °C (less than 3 s) but also high electrothermal efficiency with low power density.

From the relatively constant resistance–current behavior shown in Figure 5b, it can be observed that the LIG heaters exhibit ohmic behavior. Both the current and resistance show drastic changes when voltage is applied. They remained almost constant while the voltage was applied. As seen in Figure 5f, a step-like increase in temperature from 1 to 6 V was recorded to evaluate the stability and Joule heating performance of the LIG heater fabricated at a scanning speed of 60 mm/s, showing the best electrical properties. The test indicates six repeated cycles of maintaining the increased temperature for 30 s and then increasing the voltage applied to the heater to ensure its stability. It takes very little time to reach the saturation temperature, implying its outstanding rapid thermal change.

After studying the structural, electrical, and thermal properties of the LIG heaters, our heaters were also analyzed for electrothermal stability by monitoring the temperature change with a bending test [12,61]. Figure 6 shows the temperature variations of the heaters with a size of 100 mm^2^ during 16,000 s of bending with a bending radius of 10 mm and a total frequency of 0.5 Hz. The electric power was set to 2 W to achieve a steady temperature of 53 °C, according to the electrothermal experiments in Figure 6 on a rectangular (100 mm^2^) heater. Under these conditions, an electric power of 2 W was applied to the LIG patterns for 20 s and discontinued for another 20 s during 400 repeated cycle tests. The maximum changes in temperature reported were below 13.7 °C, resulting in a relative variation of temperature under 3%, following the periodic bending of the substrate. The cycle test showed outstanding stability and durability without any degradation under repeated bending.

### 3.4. Application of LIG Heater for De-Icing

To investigate the performance of the heater for de-icing applications, we dropped 3 mL of water on the LIG heater and froze it. Figure 7a–c shows the temperature change plot for de-icing. We applied 12 V to observe Joule heating for the thermal performance of the LIG heater when iced. Melting water was observed through Joule heating of the LIG heater. As the temperature was rapidly increased to 49 °C, the LIG heater with a scanning speed of 20 mm/s showed the best heating performance, with water melting after 60 s. The LIG heater with a scanning speed of 40 mm/s melted the water earlier than that at a scanning speed of 60 mm/s. It reached the highest temperature of 78.2 °C among the heaters. The frozen water on the LIG heater with a scanning speed of 60 mm/s gradually melted as the temperature increased from 26.3 to 61.5 °C. After 83 s, the frozen water was completely removed from the LIG heater.

As shown in Figure 7d, the contact angle of the LIG heater was analyzed depending on the laser scanning speed from 20 to 100 mm/s. The heater fabricated at a lower scanning speed displayed more hydrophilicity. This is because the surface containing more oxygen functional groups in contact with water increases [62]. Figure 7e indicates the de-icing test of our LIG heater with real images of frozen and melted water.

In addition, a de-icing experiment was conducted using two different LIG heater pads for de-icing 3 mL of ice. We applied 12 V to observe the water melting in the Kapton tape/LIG heater and Kapton tape/aluminum foil/LIG heater, as shown in Figure 8a and (b). In both samples, the ice gradually melted as the temperature increased from 26.3 to 149.2 °C. Aluminum foil was selected as the optimal substrate with an outstanding thermal conductivity. Appendix A shows that the LIG heater pad/aluminum foil shows sufficient performance to convert water into steam. To protect the LIG heater surface with high roughness, we shielded the LIG surface with Kapton tape. Attaching the Kapton tape enhances the accumulation of the heat generated during Joule heating in the large pores inside the high rough LIG heater, which causes a rapid temperature rise, as depicted in Figure 8c. With 3 mL of water frozen on the Kapton tape/LIG, the state change of water occurred through Joule heating. The surface temperature reached approximately 100 °C within a few seconds. The ice gradually melted as the temperature increased from 26.3 to 149.2 °C. After approximately 40 s, the ice was completely removed from the Kapton tape/LIG heater, as shown in Figure 8d. These two experiments indicate the LIG heater as a potential prospect for de-icing systems.

## 4. Conclusions

Although LIG has an immense potential in the electro-thermal properties for electronic devices, there is limited research on LIG-based electrothermal applications such as a wearable heat pad and de-icing applications using the electrothermal effect based on localized Joule heating. We report LIG electro-thermal heaters fabricated by a 355 nm second UV pulsed laser according to the laser scanning speed. The defective abundant and flexible LIG patterns were investigated by FE-SEM, TEM, Raman spectroscopy, TGA, XPS, XRD, BET, and contact angle measurement.

In summary, the following conclusions can be drawn from the results of this study.

(a)Our LIG heater shows excellent characteristics such as its high porosity, light weight, and small LIG pattern thickness. We adopted laser direct writing (LDW) to irradiate the substrates with computer-aided 2D CAD data for printed electronics under ambient conditions.(b)All flexible LIG heaters fabricated according to the laser scanning speed showed fast response times, reaching a high thermal temperature of 190 °C within 3 s. The LIG heater demonstrated a rapid response time, reaching equilibrium within less than 3 s, and achieving temperatures up to 190 °C using relatively low DC voltages of approximately 10 V.(c)The LIG heater pad exhibited good flexibility and durability in the bending test. The maximum reported temperature changes were below 13.7 °C, resulting in a relative variation of temperature under 3%, following the periodic bending of the substrate during 16,000 s of bending with a bending radius of 10 mm and a frequency of 0.5 Hz.(d)We applied our LIG heater pad for the purpose of de-icing to demonstrate its excellent performance.

## Figures and Tables

**Figure 1 nanomaterials-11-03093-f001:**
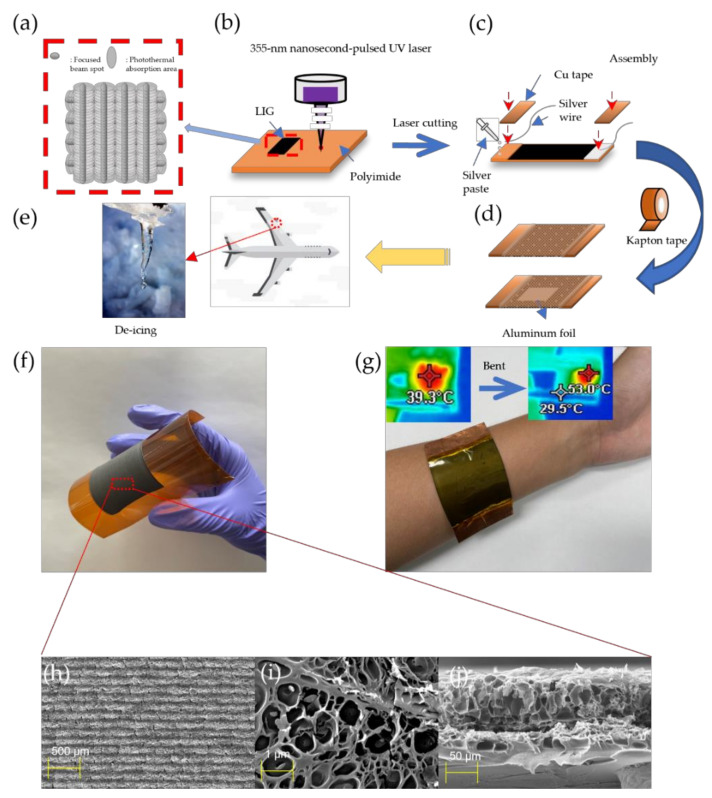
Fabrication process of proposed LIG heater: (**a**) illustration of laser pulse spot, (**b**) 355 nm pulsed UV laser irradiation onto PI:LIG heater platform (**c**) LIG heater sealed with silver paste and wire and Cu tape, (**d**) encapsulation of LIG heater with Kapton tape, (**e**) schematic illustration of LIG heater pad for de-icing (**f**) actual photograph of flexible LIG pattern, (**g**) LIG heater pad for wearable device. FE-SEM images of LIG pattern’s (**h**) top view, (**i**) interior, and (**j**) intersection.

**Figure 2 nanomaterials-11-03093-f002:**
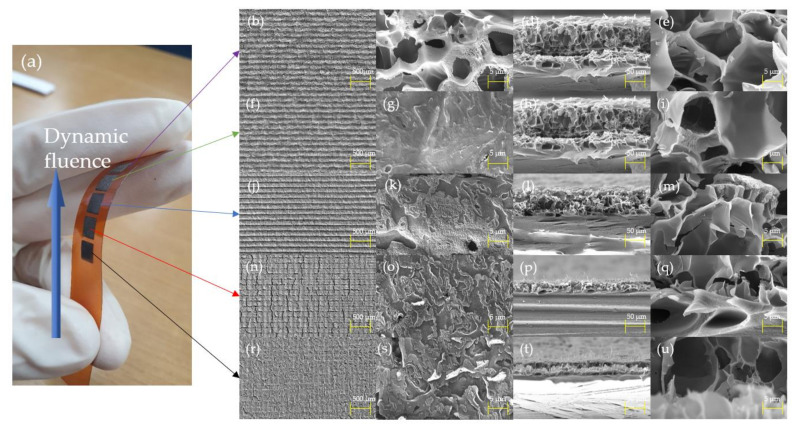
(**a**) Actual photograph of LIG patterns. FE-SEM images of LIG patterns fabricated at a laser scanning speed of (**b**–**e**) 20 mm/s, (**f**–**i**) 40 mm/s, (**j**–**m**) 60 mm/s, (**n**–**q**) 80 mm/s, and (**r**–**u**) 100 mm/s.

**Figure 3 nanomaterials-11-03093-f003:**
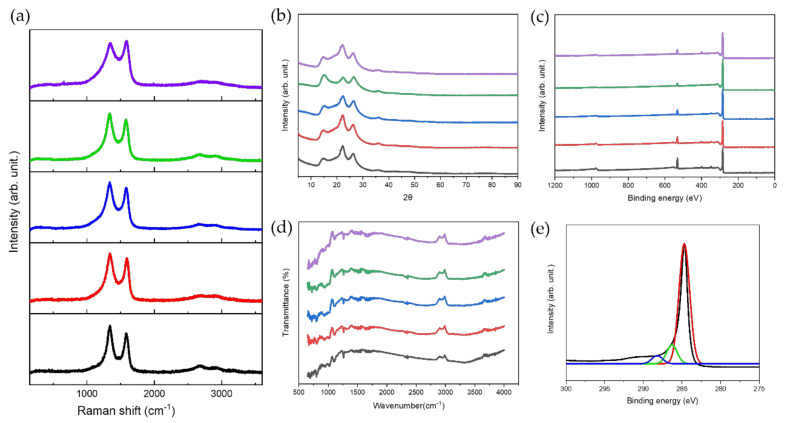
(**a**) Raman spectra of LIG patterns, (**b**) XRD results of LIG patterns, (**c**) XPS characteristics of the LIG patterns according to the laser speed, (**c**) XPS, (**d**) FTIR analysis of LIG patterns, (**e**) XPS C1s data of LIG patterns fabricated at laser speed of 60 mm/s (violet: 100 mm/s, green: 80 mm/s, blue: 60 mm/s, red: 40 mm/s black: 20 mm/s).

**Figure 4 nanomaterials-11-03093-f004:**
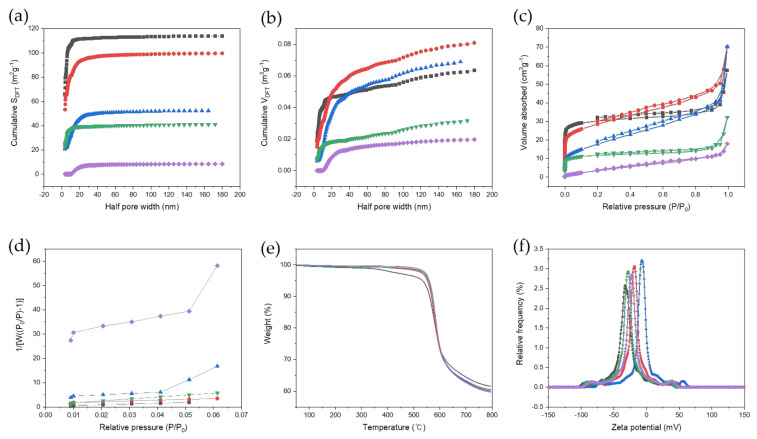
(**a**) Cumulative surface area and (**b**) cumulative pore volume derived from nitrogen physisorption data of the prepared LIG patterns fabricated according to laser scanning speed. (**c**) N2 adsorption–desorption isotherms of LIG patterns fabricated with different laser scanning speeds. (**d**) Plot of 1/[W((W(Po/P)-1)] and relative pressure for surface area calculation of LIG patterns, (**e**) TGA curves of LIG patterns, (**f**) zeta potential distribution curves for LIG patterns (violet: 100 mm/s, green: 80 mm/s, blue: 60 mm/s, red: 40 mm/s, black: 20 mm/s).

**Figure 5 nanomaterials-11-03093-f005:**
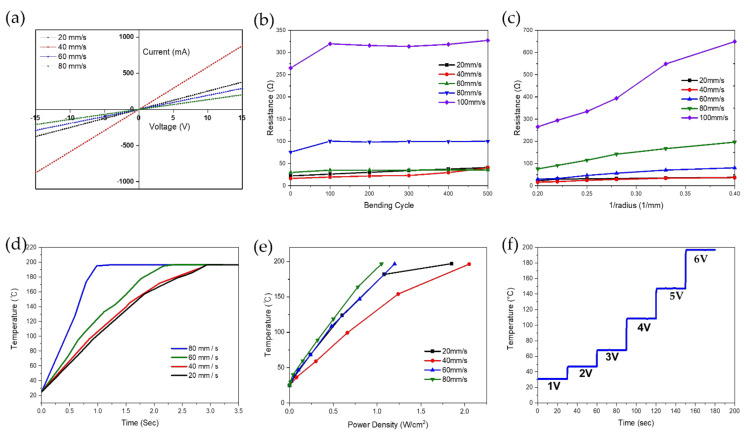
(**a**) I–V curve measurement of LIG patterns fabricated according to laser scanning speed. (**b**) Change in electrical resistance after cyclic bending. (**c**) Change in resistance as LIG patterns are bent by a cylinder of different radius. (**d**) Time-dependent characteristics of transient time to reach 200 °C under 12 V. (**e**) Temperature as a function of the applied electrical power density for LIG heaters. (**f**) Temperature evolution of LIG heaters at stepwise voltage rise from 1 to 6 V.

**Figure 6 nanomaterials-11-03093-f006:**
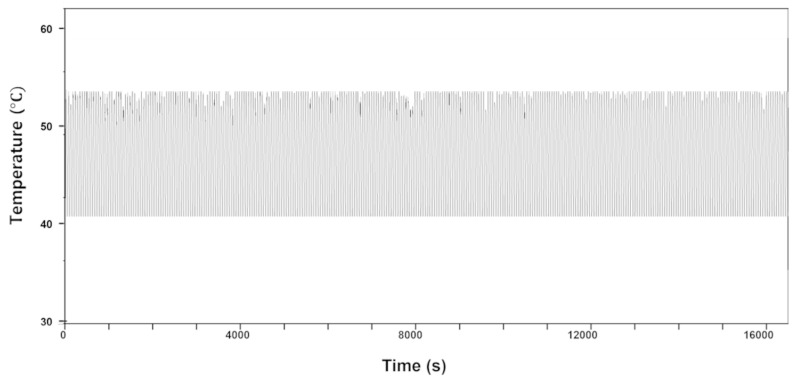
Cycle test for temperature change of LIG heater pad.

**Figure 7 nanomaterials-11-03093-f007:**
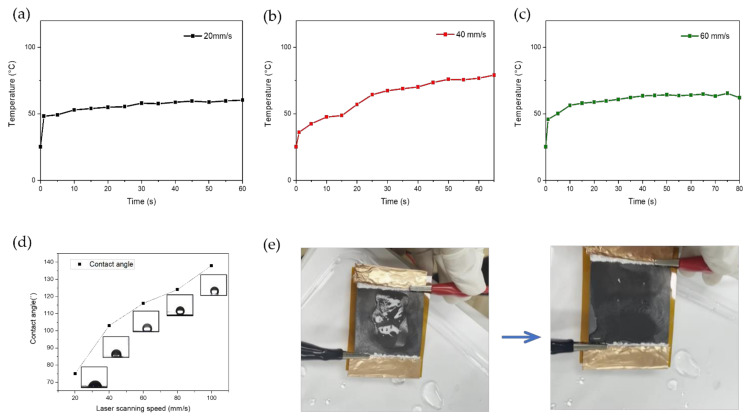
Temperature change plot for de-icing of LIG heater (50 mm × 50 mm) fabricated at a laser scanning speed of (**a**) 20 mm/s, (**b**) 40 mm/s, and (**c**) 60 mm/s during Joule heating of 12 V. (**d**) Contact angle depending on the laser scanning speed. (**e**) De-icing test of water droplet (3 mL) with LIG heater.

**Figure 8 nanomaterials-11-03093-f008:**
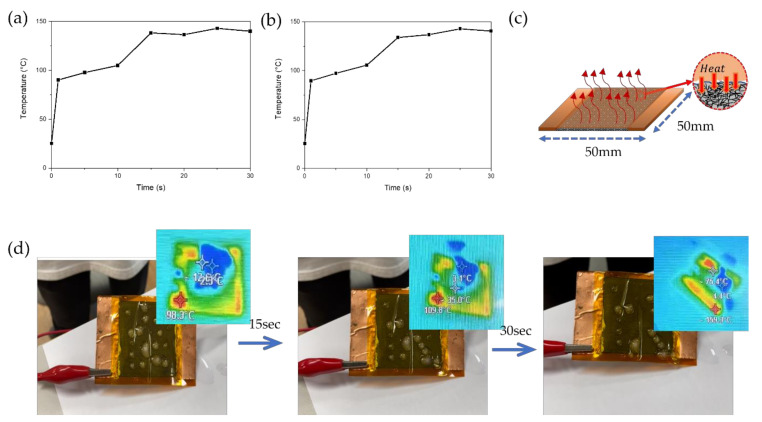
Joule heating (12 V) by attaching Kapton tape on LIG heater (50 mm × 50 mm): (**a**) de-icing test (1.5 mL) with Kapton/LIG heater, (**b**) de-icing test (1.5 mL) with Kapton/aluminum foil/LIG heater. (**c**) Schematic of LIG heater for heat accumulation in pore structure and roughness pattern. (**d**) De-icing test of water droplet (3 mL) with Kapton tape/LIG heater.

## Data Availability

The data presented in this study are available on request from the corresponding author.

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
