# Peer review of "Laser-Induced Graphene Heater Pad for De-Icing"

_nanomaterials, 2021, doi:10.3390/nano11113093_

Round 1

Reviewer 1 Report

I congratulate the authors for their work. Mainly, they tried to demonstrate that the practical translation of a recent technological material (LIG) is possible, in this case for de-icing and wearable pad heaters. They thoroughly supported their claims with sound experimental data. However, my main criticism relies on the limited novelty for this type of LIG application, as the available literature reports many approaches for LIG heaters. 

Suggestions and remarks that I think will ameliorate the manuscript can be found in the PDF file that I am uploading.

I wish the best to the authors, namely a fruitful research activity.

Author Response

Q1. my main criticism relies on the limited novelty for this type of LIG application, as the available literature reports many approaches for LIG heaters. 

A1. Recently, most of the LIG heaters are manufactured using a CO2 laser to produce 3-D carbon networks with strong heat and pressure. They show heaters with a lot of open pores to generate heat. However, this heat is quickly dispersed in all directions, so it is not suitable for defrosting.  Using our 355nm UV pulsed laser, we can fabricate unique structures with relatively little heat and pressure.

I would like to inform you the heat generation mechanism of our LIG heaters for defrost

LIG patterns fabricated at the laser speed of 100 mm/s cannot be used as a heater because there is not enough energy to from a 3-D conductive carbon network.

Overall chemistry and composition of our heaters (20 mm/s, 40 mm/s, 60 mm/s and 80 mm/s) show similar characteristics (XPS, XRD, Raman spectroscopy, and FT-IR). As the dynamic fluence increases, it can be seen that a large number of pore structures are formed as seen in FE-SEM images (Figure 2). This structure, in which there are more pores on the inside than on the surface, has potential to be used as heater for de-icing (Figure 1 (h-j)). We expect that heat generated from a large surface area of heaters fabricated at the laser speed of 20mm/s tends to accumulate in the pores and shows excellent performance in de-icing. And we simply add Kapton tape on the heater to increase the heat condensation to be more effective for de-icing. We apologize for not concisely summarizing the details. I hope my answers helps you understand a little bit.

Q2. Meanwhile, the format of the paper is messy, for example: In line 153-154 page 4, the equation is missing. Superscript and subscript formats for many expressions are missing. BET is not defined in line 159.

A2. By your checking, the descript has been corrected. The equations (in line 153-154, page 4) have been removed. And I corrected abbreviation BET. I have attached the modified file, so please take a look.

Q3. The interpretation for Fig. 6 is confusing. Is the curve recorded with a periodic bending, or a repeated cycle of applied electric power, or both? The inset indicates a temperature increment from 39.3 to 53 degree caused by bent. Is that true? Anyway, the figure and examination of this experiment is confusing.

A3. Figure 6 is the recorded curve of a repeated cycle of applied electric power of 2W applied to our LIG heater. As mentioned in the paper, the electric power was applied to the heater for 20 seconds, reaching 53 degree, and not applied for another 20 seconds, decreasing the temperature to 40.9 degree. The cycle was repeated for 400 times which took about 16,000 seconds for our test to be finished. It shows the outstanding stability and durability of the heater without any degradation during the cycle test. I deleted inset figure because I thought it was confusing. Thanks for the good point.

And I modified it according to the pdf file you attached. Thanks for checking in detail.

In 43 lines page 1, sp3-hybridized is modified.

In 57 lines page 2, I modified thermoelectric to electrothermal.

In 89 lines page 2, I deleted the lines you underlined.

In lines 106-107 page 3, I deleted the lines as you pointed out.

In lines 118 page 3, I corrected aircraft to aircraft’s wings

In lines 123 page 3, I corrected pastern to pattern.

In lines 145 page 3, I corrected CA to contact angle.

In lines 152 page 4, I deleted 153-154 lines and added reference to as you pointed out (156,159 lines).

In lines 166 page 5, Sorry for the confusing to you, attached over the heater is Kapton tape. The substrate is PI film.

In lines 167 page 5, I corrected f to g.

In lines 188 page 6, Laser induced plasma (LIP) was written down, but it was deleted due to confusion in the contents.

In lines 200 page 6 and 213 page 7, As you said, the content was not delivered well, so I edited it.

In lines 219 page 7, I corrected graphene oxide to reduced graphene oxide.

In lines 293 page 9, I corrected absorption to adsorption.

In lines 209 page 9, I corrected at to for.

In lines 313 page 10, the resistance is corrected to electrical characteristics.

In lines 316 page 10, I corrected the formatting.

In lines 320 page 10, I added “under 12 V” and in the Figure 5 (d) caption.

In lines 323 page 10, I deleted 0-7 V

In lines 331 page 10, I corrected when to while.

In lines 345 page 11, I corrected 10x10 and formatting.

In lines 375 page 12, I added each labels on the plot (Figure 7. (a, b, and c)

In lines 397 page 13, sorry for not adding color bar next to each one. I also want to do what you said, but I can't insert a bar next to it due to a problem with the equipment.

In liens 400-401 page 13, I corrected thermoelectric to electrothermal.

In lines 408 page 13, I deleted the lines.

In lines 412, I corrected “can be used in industry” to shows excellent.

In liens 427-428 page 14, I deleted the lines.

Thanks to your review, the quality of the thesis has been improved.

Thanks for the review.

Best regards,

JUNUK LEE

Reviewer 2 Report

The paper reports a preparation of LIG heaters and investigates their structure and heating performance with DC bias. The topic seems interesting and have potential in many applications. Although the paper presents detailed characterizations, and electrical and thermal properties, and de-icing application, the underlying mechanism between structure and heating performance is not clear.

For instance, the 80 mm/s sample has largest resistance and temperature increment performance, while the de-icing experiments say the 20 mm/s sample shows the best heating performance. Then, what is the connection of the de-icing performance with the electrical properties, as well as with the structural features. Although the de-icing application is interesting, I think the paper should be revised to demonstrate the reasonable mechanism for the heating performance related to the structure and properties.    

Meanwhile, the format of the paper is messy, for example:

In line 153-154 page 4, the equation is missing.

Superscript and subscript formats for many expressions are missing.

BET is not defined in line 159.

The interpretation for Fig. 6 is confusing. Is the curve recorded with a periodic bending, or a repeated cycle of applied electric power, or both? The inset indicates a temperature increment from 39.3 to 53 degree caused by bent. Is that true? Anyway, the figure and examination of this experiment is confusing.

Author Response

Q1. the paper reports a preparation of LIG heaters and investigates their structure and heating performance with DC bias. The topic seems interesting and have potential in many applications. Although the paper presents detailed characterizations, and electrical and thermal properties, and de-icing application, the underlying mechanism between structure and heating performance is not clear.

A1. I would like to inform you that LIG patterns fabricated at the laser speed of 100 mm/s cannot be used as a heater because there is not enough energy to from a 3-D conductive carbon network.

Overall chemistry and composition of our heaters (20 mm/s, 40 mm/s, 60 mm/s and 80 mm/s) show similar characteristics (XPS, XRD, Raman spectroscopy, and FT-IR). As the dynamic fluence increases, it can be seen that a large number of pore structures are formed as seen in FE-SEM images (Figure 2). This structure, in which there are more pores on the inside than on the surface, has potential to be used as heater for de-icing (Figure 1 (h-j)). We expect that heat generated from a large surface area of heaters fabricated at the laser speed of 20mm/s tends to accumulate in the pores and shows excellent performance in de-icing. And we simply add Kapton tape on the heater to increase the heat condensation to be more effective for de-icing. We apologize for not concisely summarizing the details. I hope my answers helps you understand a little bit.

Q2. Meanwhile, the format of the paper is messy, for example: In line 153-154 page 4, the equation is missing. Superscript and subscript formats for many expressions are missing. BET is not defined in line 159.

A2. By your checking, the descript has been corrected. The equations (in line 153-154, page 4) have been removed. And I corrected abbreviation BET. I have attached the modified file, so please take a look.

Q3. The interpretation for Fig. 6 is confusing. Is the curve recorded with a periodic bending, or a repeated cycle of applied electric power, or both? The inset indicates a temperature increment from 39.3 to 53 degree caused by bent. Is that true? Anyway, the figure and examination of this experiment is confusing.

A3. Figure 6 is the recorded curve of a repeated cycle of applied electric power of 2W applied to our LIG heater. As mentioned in the paper, the electric power was applied to the heater for 20 seconds, reaching 53 degree, and not applied for another 20 seconds, decreasing the temperature to 40.9 degree. The cycle was repeated for 400 times which took about 16,000 seconds for our test to be finished. It shows the outstanding stability and durability of the heater without any degradation during the cycle test. I deleted inset figure because I thought it was confusing. Thanks for the good point.

Thanks to your review, the quality of the thesis has been improved.

Thanks for the review.

Best regards,

JUNUK LEE

Round 2

Reviewer 2 Report

The revised manuscript has addressed the comments and I think the paper could be accepted for publication.